# Reliability of Instantaneous Wave-Free Ratio (iFR) for the Evaluation of Left Main Coronary Artery Lesions

**DOI:** 10.3390/jcm8081143

**Published:** 2019-07-31

**Authors:** Salvatore De Rosa, Alberto Polimeni, Giovanni De Velli, Micaela Conte, Sabato Sorrentino, Carmen Spaccarotella, Annalisa Mongiardo, Jolanda Sabatino, Marco Contarini, Daniel Todaro, Ciro Indolfi

**Affiliations:** 1Division of Cardiology, Department of Medical and Surgical Sciences, Magna Graecia University, 88100 Catanzaro, Italy; 2Presidio Ospedaliero Umberto I, 96100 Siracusa, Italy; 3S.M. Goretti Hospital, 04100 Latina, Italy; 4URT-CNR, Magna Graecia University, 88100 Catanzaro, Italy

**Keywords:** instantaneous wave-free ratio (iFR), fractional flow reserve (FFR), left main coronary artery (LMCA), coronary stenosis

## Abstract

The assessment of the left main coronary artery (LMCA) by coronary angiography has several limitations. The fractional flow reserve (FFR) is useful for the functional evaluation of LMCA stenoses. The instantaneous wave-free ratio (iFR), a resting index, was developed to simplify functional coronary assessment. However, its performance for LMCA stenoses has yet to be explored. The iFR was measured at rest, and the FFR was measured under maximal hyperemia. We calculated that a sample size of 90 lesions would have provided 90% power at a 5% significance level to detect an Area Under the Curve (AUC) < 0.7 for the iFR to identify FFR-positive stenoses. A total of 91 measurements were performed on angiographically intermediate LMCA stenoses at three centers. The comparison between the iFR and the FFR showed a significant correlation (r = 0.67, *p* < 0.001). At receiver operating characteristic (ROC) analysis, the iFR revealed a good diagnostic performance when compared to the FFR (AUC = 0.84; *p* < 0.001). A classification agreement between the iFR and the FFR was recorded in 81% of cases. The left ventricular ejection fraction (LVEF) was an independent predictor of the discrepancy between the FFR and iFR values (*p* = 0.040). The present study is the first demonstrating that the assessment of LMCA stenoses with the instantaneous wave-free ratio is a reliable adenosine-free alternative to classic fractional flow reserve. If confirmed in larger populations, these findings could be of relevance for real world daily practice.

## 1. Introduction

The quantitative evaluation of coronary angiography has limitations. More so with intermediate stenoses [1,2]. Quantification is more difficult for stenoses of the left main coronary artery (LMCA), with an agreement in stenosis grading among experienced interventional cardiologists ranging from 41% to 59% [3,4,5]. In fact, the short length, the frequent overlapping with side branches, the lack of a reference segment, the sometimes unavoidable “contrast streaming” at the ostium due to difficult catheter positioning, and the singular reverse tapering, such that its caliber often increases from the ostium to the distal section, make a precise angiographic evaluation of LMCA disease very difficult, even for experienced cardiologists [6].

The fractional flow reserve (FFR), calculated as the distal to proximal (aortic) pressure ratio during hyperemia, is currently the gold standard for the functional assessment of coronary stenoses’ severity, as its use has been associated with a clinical benefit in several studies [7,8,9]. 

More recently, the use of the FFR for the evaluation of LMCA stenoses has been tested in several studies, with promising results [10,11,12,13,14,15,16,17]. On the other hand, some issues are still limiting use of the physiologic assessment of the LMCA. The lack of randomized studies evaluating long-term prognosis is an important limitation. Moreover, the frequent involvement of LMCA bifurcation often requires the measurement of both distal daughter branches. Finally, the not infrequent presence of additional stenoses of downward vessels may have an impact on pressure measurements.

More recently, the instantaneous wave-free ratio (iFR) was introduced as an alternative, adenosine-free index for the functional evaluation of coronary stenoses [18,19,20,21,22]. The iFR presents some differences compared to the FFR that could potentially simplify the use of intracoronary physiology measurements. This could be particularly useful when a comprehensive evaluation of an LMCA disease is needed, such as in case of bifurcation stenoses, where multiple measurements are often performed in both daughter branches.

Accordingly, the aim of the present study was to test the diagnostic performance of the instantaneous wave-free ratio (iFR) for the quantitative evaluation of LMCA stenoses, compared to the FFR.

## 2. Methods

### 2.1. Patient Population

Consecutive patients undergoing a functional evaluation of the LMCA at 3 centers were included. The protocol of the study was approved by the Ethics Review Board, and all patients gave their written informed consent to the study.

### 2.2. Coronary Angiography and Intracoronary Pressure Measurement

The iFR measurement was performed after the administration of an intracoronary bolus of nitroglycerin. The pressure wire (VERRATA, VOLCANO corporation, San Diego, CA, USA) was equalized before advancement through the target lesion. Intravenous adenosine (140 µg/kg/min) was used to measure the FFR. After every measurement, the pressure wire was pulled back to exclude a pressure drift. A wave-form analysis was mandatory before completing the examination, to avoid the presence of pressure dumping or any artifact. There was no operators’ selection in this study. All pressure measurements performed by trained personnel within the study period were included in the study.

### 2.3. Statistical Analysis

Comparisons were tested using the Pearson’s chi-square test for categorical variables. Based on the diagnostic accuracy measured for the iFR in previous studies [18,19,23,24], we calculated that a sample size of 90 lesions would have provided 90% power at the 5% significance level to detect an Area Under the Curve (AUC) < 0.7 for the iFR to identify FFR-positive stenoses. Receiver operating characteristic (ROC) curves were used to evaluate the diagnostic performance of the iFR in identifying a positive FFR measurement by the area under the curve [25]. A correction of the iFR values by the left ventricular ejection fraction (LVEF) (_corr_iFR) was obtained by dividing the measured iFR value by the base-10 logarithm of the LVEF. The established diagnostic cutoffs for the iFR and the FFR of, respectively, 0.89 and 0.80 were primarily used. The iFR diagnostic cutoff corresponding to the FFR cutoff of 0.75, and the diagnostic cutoff for _corr_iFR as set at the maximum value of the Youden index. A univariable regression analysis was used to test variables affecting the numerical difference between the iFR and FFR measurements, as well as the classification mismatch. A generalized linear model was used for multivariable analysis to test for independent predictors of the iFR–FFR discrepancy. Predictors to be tested at multivariable analysis were selected if significantly correlated to the degree of the iFR–FFR discrepancy (LVEF and hypertension). In addition, diabetes was added, as it is known to be associated with a high prevalence of microvascular myocardial disease.

The net reclassification improvement (NRI) was calculated by assigning a +1 to lesions that were correctly reclassified using the _corr_iFR respect to the standard iFR and by assigning a −1 to lesions that were incorrectly reclassified using the _corr_iFR respect to the standard iFR. Lesions not reclassified were assigned a 0. The scores in each group (FFR positive and FFR negative) were divided by the number of subjects in that group. Finally, the NRI was calculated as the sum of the values obtained for each of these two groups.

A V-shaped plot (V-plot) was calculated as described elsewhere [26] to display the classification agreement (CA) across different ranges of FFR values. *p* < 0.05 was considered statistically significant.

To compare the different Impacts of a Downstream Stenosis on the FFR (IDS-FFR) and the iFR (IDS-iFR), the calculated differences were normalized by the respective diagnostic cutoff as follows: IDS-FFR = [FFR_vessel with downward stenosis_ − FFR_vessel with downward stenosis_]/0.80 and IDS-iFR = [iFR_vessel with downward stenosis_ − iFR_vessel with downward stenosis_]/0.89. All analyses were performed with SPSS software, version 23.0 (SPSS/IBM, Armonk, NY, USA).

## 3. Results

### 3.1. Baseline Characteristics

A total of 91 measurements of LMCA lesions were included in this multicenter study, performed in 80 patients of 65.7 ± 9.6 years of age. Participants were mostly males (83.8%). Detailed patients’ characteristics are reported in Table 1. 

The most frequent stenosis location was the LMCA bifurcation (70.3%), followed by ostial lesions (18.7%). The LMCA shaft was the site of the intermediate stenosis in the remaining 11% of measurements. Most measurements were performed with the pressure wire advanced into the left anterior descending artery (LAD) (74.4%).

Double measurements were performed in eight procedures to independently assess the pressure gradients in the left circumflex (LCx) and the LAD. Key procedural characteristics are reported in Table 2.

### 3.2. Diagnostic Characteristics

The iFR was found to correlate with the FFR (r = 0.67, *p* < 0.001) (Figure 1), as shown by the scatterplot depicting the correlation between the instantaneous wave-free ratio (iFR) and the fractional flow reserve (FFR).

At ROC analysis, the iFR showed an excellent diagnostic performance to identify significant stenoses using the established FFR cutoff of 0.8 (AUC = 0.84; *p* < 0.001) (Figure 2A), resulting in a sensitivity of 80% and a specificity of 78%. Using the established cutoffs of 0.8 for the FFR and 0.89 for the iFR (≤0.89 classified as positive), we found concordance between the outcome of the iFR and the FFR in 81% of measurements. Among discordant measurements (19%), in nine cases, the iFR results were negative, while FFR results were positive. In the remaining eight tests, the iFR results were positive, but the FFR results were negative.

Diagnostic performance was analyzed in patients’ subgroups with stable coronary artery disease (CAD) (*n* = 38) or acute coronary syndrome (ACS) (*n* = 42). As shown in Figure 3, a similar diagnostic performance was observed between stable CAD (left panel) and ACS patients (right panel), with no significant difference between the AUCs (*p* = 0.366). In line with these results, a diagnostic agreement was comparable between stable CAD (79%) and ACS (84%) patients.

The V-plot displayed in Figure 2B is a sample-independent methodology to report accuracy across the spectrum of disease severity confirmed a very good classification agreement (CA) between the indices, with a focal fall within the so-called “grey-zone” of FFR measurement (FFR between 0.77 and 0.79). The CA between the iFR and the FFR was lowest (CA = 54%) within the 0.77–0.79 FFR range, and it was significantly higher outside in the remaining lesions’ set (CA = 89%, *p* < 0.001).

At univariable analysis, the left ventricular ejection fraction (LVEF) was inversely correlated with the degree of numerical discrepancy between the iFR and the FFR measurements (r = −0.257; *p* = 0.017): Larger iFR–FFR differences were found for lower LVEF values (Figure 4A). At multivariable analysis, after the addition of hypertension and diabetes (potential influencing factors), the LVEF remained the only independent predictor of the iFR–FFR discrepancy (*p* = 0.040) (Table 3). 

Accordingly, the correction of the iFR for the LVEF (_corr_iFR) resulted in an increase of the diagnostic performance of the iFR at ROC analysis (AUC = 0.87; *p* < 0.001) (Figure 4B). The use of _corr_iFR yielded an NRI of 0.1, compared to the standard iFR.

When the FFR cutoff of 0.75 was applied, the diagnostic performance of the iFR was better (AUC = 0.91; *p* < 0.001) compared to the 0.80 cutoff, resulting in a sensitivity of 83% and a specificity of 86%. Using the 0.75 cutoff for the FFR, we found concordance in 84% of measurements.

In line with previous results from Fearon et al. [27], the presence of a further downward stenosis in the measured vessel had an impact on the measured FFR value. In particular, the mean difference between FFR values measured in the daughter branch with the downwards stenosis and the other branch with no stenosis was 0.18 ± 0.05. The mean difference between the iFR values measured in the daughter branch with the downwards stenosis and the other branch with no stenosis was 0.08 ± 0.02. The presence of a downward stenosis had a significantly larger impact on the FFR (IDS-FFR = 0.14 ± 0.06) than on the iFR (IDS-iFR = 0.10 ± 0.05; *p* = 0.042).

## 4. Discussion

Our results demonstrate that (1) the measurement of iFR is reliable in patients with LMCA disease; (2) the iFR has good correlation and classification match with FFR; (3) the diagnostic performance of the iFR is similar in stable CAD or ACS patients; (4) the adjustment of iFR results by the LVEF improves its diagnostic performance to identify FFR-positive stenoses; (5) the presence of a further stenosis downstream of the target intermediate stenosis has an impact on both pressure indices. Despite the comparison being based on a limited number of cases, the iFR seems to be less influenced by downstream stenoses.

These findings might have an impact on the propensity to use physiological assessment in patients with LMCA stenoses. In fact, the functional assessment of stenosis severity by means of pressure-derived indices is a clinically useful and meaningful diagnostic tool to identify ischemia-inducing stenoses [28,29]. As demonstrated by the V-plot in Figure 2B, diagnostic agreement between the iFR and the FFR is very high, with the largest disagreement in a narrow interval around the diagnostic FFR cutoff. This latter finding is quite common when a discrete cutoff is applied to continuous variables, such as pressure indices. The positive impact of physiological assessment on the LMCA Percutaneous Coronary Intervention (PCI) was further supported by the results of the recent “Percutaneous coronary angioplasty versus coronary artery bypass grafting in treatment of unprotected left main stenosis” (EXCEL) and “Percutaneous coronary angioplasty versus coronary artery bypass grafting in treatment of unprotected left main stenosis” (NOBLE) trials [30]. In fact, these studies that incorporated the FFR to guide PCI in a relevant percentage of procedures strongly contributed to the change of the cumulative evidence around the revascularization of LMCA disease toward a positive outlook for PCI [30]. However, since the FUTURE trial (NCT01881555) included 11% of patients with LMCA stenosis, their results of excess mortality in the FFR-guided arm shed some shadows on clinical reliability of coronary physiology. On the contrary, the most recent SYNTAX II trial showed that a contemporary management strategy including the iFR and the FFR is associated with excellent results after PCI [31].

Despite its recognized clinical utility, the use of physiology guidance is currently low worldwide, more so for stenoses of the LMCA [9,10,11,12]. In this panorama, the use of an adenosine-free pressure index might be useful to guide revascularization of coronary disease involving the LMCA. In fact, the iFR could simplify the functional assessment of stenosis severity in the LMCA, usually more demanding than in other coronary segments. For example, the careful disengagement of the guiding catheter from the LMCA ostium is of key importance to prevent the pressure dampening and underestimation of distal pressure, making intracoronary adenosine administration particularly challenging. Care should be used in case of significant downstream coronary arterial disease, because distal stenoses have an impact on measured values [27,32]. In this regard, our findings of a discrepancy between measurements performed with the pressure wire alternatively positioned in either daughter branch are in line with previous reports [27,32]. Moreover, the preliminary finding that the presence of a downward stenosis had a significantly lower impact on the iFR than on the FFR is worth further investigation. When a downstream stenosis is present, a useful strategy can be to repeat the measurement after putting the pressure wire into the other, non-diseased, daughter vessel. Nevertheless, a pullback recording should be preferentially performed to confirm the localization of the functionally significant stenosis within the LMCA and exclude the influence by visually underestimated downstream stenoses [33].

### Study Limitations

This study was not powered to validate an iFR diagnostic cutoff for the LMCA—this was beyond the study’s aims. Nevertheless, this study provides the first evidence on the reliability of the iFR for LMCA stenosis. For the same reason, our study did not aim to evaluate the clinical outcome of an iFR-guided PCI strategy in patients with LMCA stenosis. To this regard, the recently started iLITRO study (NCT03767621) will provide additional information on the diagnostic concordance between the iFR and the FFR in patients with intermediate LMCA stenoses, along with short-term clinical outcomes.

For similar reasons, as our main focus was the evaluation of the feasibility of the iFR assessment for intermediate LMCA stenoses, we did not calculate the anatomical nor the functional SYNTAX. In fact, we did not aim at assessing the impact of the iFR on clinical decision making.

Distal pressure measurements in case of downward stenoses were not systematically undertaken. For this reason, the impact of single stenosis on trans-stenotic gradients in case of serial lesions are not available.

## 5. Conclusions

Our study is the first to demonstrate that iFR measurement is a reliable method to assess left main coronary artery stenosis severity. The iFR and the FFR showed an excellent classification agreement, with most disagreement occurring within a narrow FFR range falling within the so-called grey-zone.

This is the largest and more extensive report on reliability of the iFR for the assessment of LMCA stenoses. In fact, the recent “Use of the Instantaneous Wave-free Ratio or Fractional Flow Reserve in PCI” (DEFINE-FLAIR) trial excluded LMCA stenoses per protocol [20], while in the “Instantaneous Wave-free Ratio versus Fractional Flow Reserve to Guide PCI” (iFR-SWEEDHEART) trial, only 1.5% of measurements were performed on the LMCA [21].

## Figures and Tables

**Figure 1 jcm-08-01143-f001:**
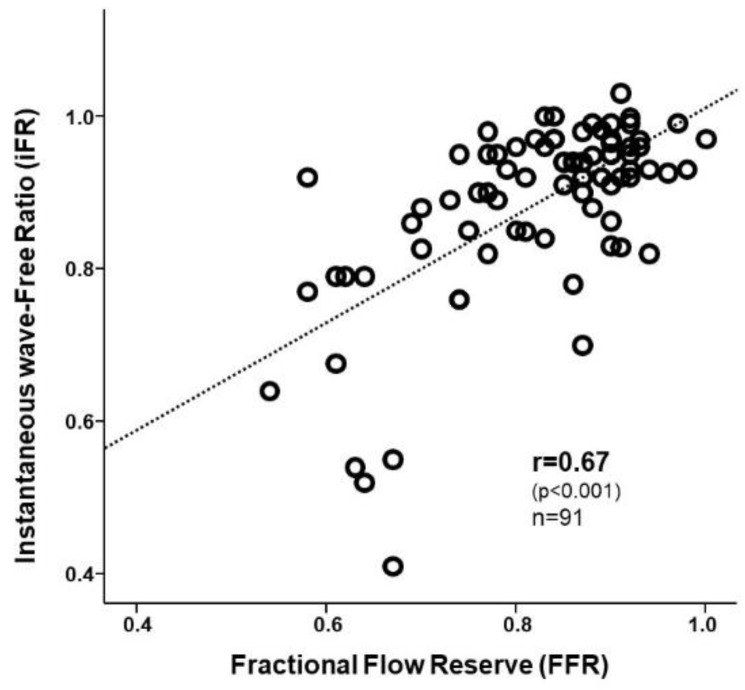
Correlation between the instantaneous wave-free ratio (iFR) and the fractional flow reserve (FFR).

**Figure 2 jcm-08-01143-f002:**
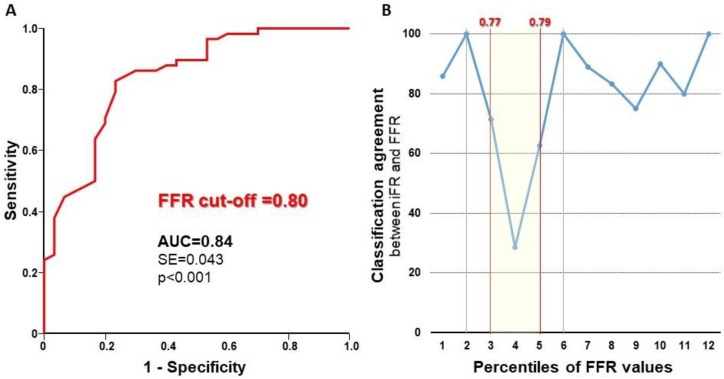
Diagnostic performance of the iFR. (**Panel A**) Receiver operating characteristic (ROC) curve analysis demonstrates an excellent performance of the iFR to detect FFR-positive stenoses. (**Panel B**) V-plot (V-shaped plot) depicting the classification agreement between the iFR and the FFR across the percentiles of FFR values measured in the study.

**Figure 3 jcm-08-01143-f003:**
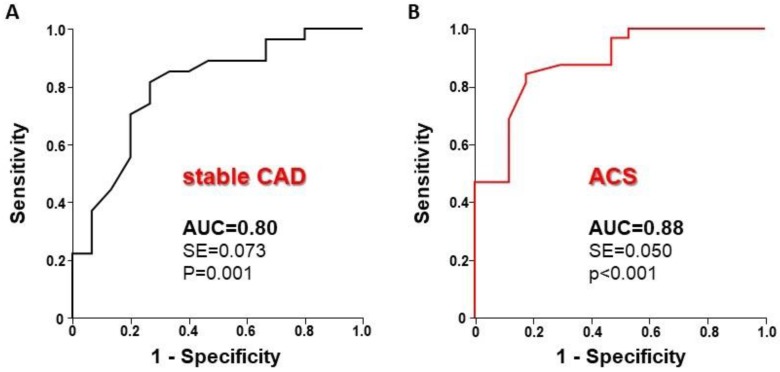
Diagnostic performance of the iFR. (**Panel A**) ROC curve analysis demonstrates an excellent performance of the iFR to detect FFR-positive stenoses. (**Panel B**) V-plot depicting the classification agreement between the iFR and the FFR across the percentiles of FFR values measured in the study.

**Figure 4 jcm-08-01143-f004:**
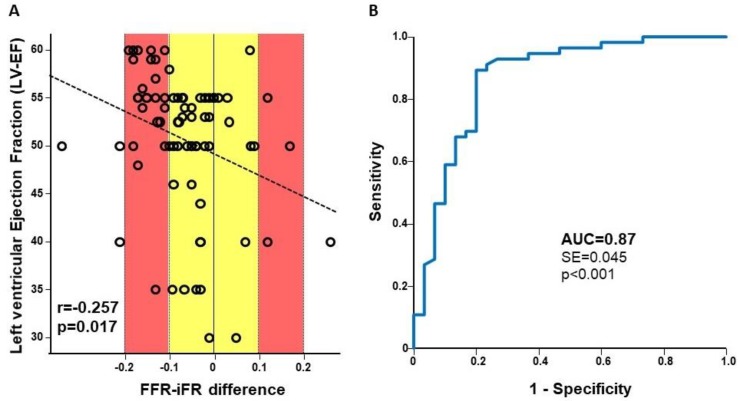
Impact of left ventricular function on the iFR. (**Panel A**) The discrepancy between measured values of the FFR and the iFR is inversely correlated to the left ventricular ejection fraction (LVEF). (**Panel B.**) The correction of iFR measurement by LVEF improves its diagnostic performance.

**Table 1 jcm-08-01143-t001:** Patients’ characteristics.

**Clinical Profile**	
Age (years)	65.7 ± 9.6
Sex (male/female)	67/13
Heart rate (bpm)	73.3 ± 13.3
BP (systolic, mmHg)	129.0 ± 29.0
BP (diastolic, mmHg)	77.1 ± 18.5
LV-EF (%)	50.5 ± 7.4
Hypertension	84.6%
Smoker	29.5%
Diabetes	23.1%
Hypercholesterolemia	44.9%
**Pharmacological Therapy**	
ASA	100%
DAPT	97%
ACE-I	79%
ARB	10%
CCB	11%
βB	93%
OHA	10%
Promethazine	0%
Ondasetron	0%

Abbreviations: BP = Blood pressure; ACS = Acute coronary syndrome; CAD = Coronary artery disease; LV-EF = Left ventricular ejection fraction; ASA = Acetylsalicylic acid; DAPT = Dual anti-platelet treatment; ACE-I = Angiotensin converting enzyme inhibitors; ARB = Angiotensin receptor blockers; CCB = Calcium channel blockers; βB = Beta-blockers; OHA = Oral hypoglycemic agents.

**Table 2 jcm-08-01143-t002:** Procedural characteristics.

**Clinical Indication**	
Stable Angina	38 (47.5%)
Unstable Angina/NSTEMI	35 (43.7%)
STEMI	7 (8.8%)
**Lesion Location**	
Ostial LMCA	17(18.7%)
LMCA shaft	10 (11%)
LMCA bifurcation	64 (70.3%)
**Pressure Wire Positioning**	
LAD	68 (74.7%)
LCx	22 (24.2%)
RIM	1 (1.1%)

Abbreviations: BP = Blood Pressure; LV-EF = Left Ventricular Ejection Fraction.

**Table 3 jcm-08-01143-t003:** Independent predictors of the iFR–FFR discrepancy.

Predictor Variable	Wald Coefficient	*p* Value
Hypertension	0.686	0.407
Diabetes	0.372	0.220
LVEF	4.216	0.040

Abbreviations: LVEF = Left Ventricular Ejection Fraction.

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
