# Peer review of "Reliability of Instantaneous Wave-Free Ratio (iFR) for the Evaluation of Left Main Coronary Artery Lesions"

_jcm, 2019, doi:10.3390/jcm8081143_

Round 1
Reviewer 1 Report
In this paper, authors underwent to test the assessment of instantaneous flow reserve of intermediate stenoses of the left main and to determine its concordance with classic fractional flow reserve (FFR). They found that iFR performed well compared to FFR reaching AUC of 0.84 while the concordance of FFR and iFR was in 81% of cases. LVEF was identified as an independent predictor of FFR/iFR mismatch.
Relevant ethical disclosures are present and statistical methods are generally appropriate with some details that would need further attention (See comments below).
General comments:
This is a relevant study that provides practice-related findings on methods that are increasingly being used to assess functional stenoses in interventional cardiology. At this moment data on IFR use in left main stenoses are lacking and are mostly extrapolated from trials that were performed on other lesions to a far larger extent, therefore, I think that this study holds value in that regard.
My main concern regarding this manuscript is a very lacking characterization of the patient population that was enrolled, except some basic info in Results (section 3.1.).
I would strongly advise authors to prepare and provide Table 1. with a display of baseline characteristics of patients that were enrolled in this study with all relevant data on baseline pharmacotherapy and PCI indication. We do not know anything about these patients regarding the setting in which they were enrolled.
Were these patients with stable angina, with the acute coronary syndrome and if so, which percentage of these were acute cases and which ones were elective? We lack data on study setting so authors would need to provide such information. For example, SWEDEHEART enrolled patients with stable angina and acute coronary syndrome while DEFINE-FLAIR enrolled patients with CAD...etc., this needs to be discussed in light of your population.
I would be curious to see if the iFR/FFR correlation differs between ACS and stable CAD patients and if the classification match of IFR in ACS was inferior to stable CAD? Please provide such analysis if feasible.
Distal pressure measurements were not systematically undertaken by following the same methodology which could have influenced results. While I understand that advancement and apposition of pressure wire in the same manner/pathway is not always feasible in the cath lab, this should be acknowledged as one of the limitations of this study. We also have no data on concomitant stenoses in daughter branches etc which can significantly skew pressure results. We lack data on coronary data, SYNTAX scores, etc. this should be discussed as a limitation.
Results of Figure 2b show that the largest disagreement between iFR and FFR was in „0.77-0.79“ area which is somewhat expected and has been problematic in previous studies as well. Whereas the graph nicely shows decent concordance in other spectrums
This study showed it is feasible to use IFR in the determination of functional stenosis of the left main. This has practical implication since iFR has some advantages over FFR and the fact that it does not require adenosine is important, especially from patients and side-effect point of view and also decreases cath lab expenditures on adenosine shots.
Results of univariable regression analysis are not clear. Authors should elaborate on which variables they decided to incorporate into the univariable model and why did they choose these particular variables? In this way, it leaves an impression as if they were chosen randomly. Also, why was the multivariate model not adjusted for sex, age, eGFR, and some other potentially relevant covariates? Authors should provide their reasoning on this and also clearly define their models in the Methods section.
I like the fact that authors tackled the issue of IFR feasibility over FFR in LM lesions exclusively and this is to be commended, however, ongoing works of other groups should be mentioned as well in this setting, and I would advise authors to briefly shed their light on those as a future and ongoing studies. For example, include an iLITRO study that is ongoing (NCT03767621) that assesses the concordance of FFR and iFR in the intermediate lesions of LMC.
Minor comments
Line 24...seems like the word is missing here, please reformulate as „...agreement between iFR and FFR was achieved in 81% of cases“
Check all references and adhere them to journal style.
Reviewer 2 Report
This paper describes a cohort study comparing FFR and iFR in assessing LMCA compromise. IT adds to the body of evidence supporting iFR as an alternative to FFR. The authors do not describe the interventions performed, or any clinical outcomes.
The manuscript is well written and follows the author guidelines.
Originality/Novelty: The association of discrepancy with depressed LVEF is interesting. However, the fact that a downstream stenosis would affect accuracy of the measurements is not novel, as it is not possible to apply routine fluid dynamic laws in these situations (this principle is also well known in the field of echocardiography).
Significance: This adds to the body of evidence supporting physiologic coronary testing, and may lend impetus to developing the program in some institutions.
Quality of Presentation: The article is easy to read. There are some grammatical errors and poor choice of words at some places which can be easily corrected by an editor.
Scientific Soundness and Additional Information: While the authors talk about safety and feasibility, no measures of these are actually reported in the results. Were all faculty members doing the iFR or only a select few who had been trained for the purpose of this study? Were the FFR and iFR done by the same interventional cardiologist, and what order were they done in? Were they aware of the result of the first test before they started doing the other test? How much time did doing iFR add to the case? Where there any complications or problems during the measurements? How did the findings change the clinical intervention done for these patients?
Interest to the Readers: Physiologic coronary testing is a topic of great interest currently.
English Level: Average. Would benefit from editor to replace some words which sound hyperbolic.
Additional minor points:
Line 72 – “Comparison were TESTED…”
Line 131 – “LVEF” instead of LFEF
Line 155 – The study does not really establish feasibility. It is uncertain how much extra time or training would be needed to incorporate this into standard practice.
Line 159 - grammar check
Round 2
Reviewer 1 Report
Authors have addressed all my concerns effectively. I would advise this paper for publication in its present form.
Reviewer 2 Report
Thank you very much for the revisions. My comments have been sufficiently addressed.